# Fiber Orientation Estimation from X-ray Dark Field Images of Fiber Reinforced Polymers Using Constrained Spherical Deconvolution

**DOI:** 10.3390/polym15132887

**Published:** 2023-06-29

**Authors:** Ben Huyge, Jonathan Sanctorum, Ben Jeurissen, Jan De Beenhouwer, Jan Sijbers

**Affiliations:** 1imec—Vision Lab, Department of Physics, University of Antwerp, 2000 Antwerp, Belgium; jonathan.sanctorum@uantwerpen.be (J.S.); ben.jeurissen@uantwerpen.be (B.J.); jan.debeenhouwer@uantwerpen.be (J.D.B.); jan.sijbers@uantwerpen.be (J.S.); 2DynXlab: Center for 4D Quantitative X-ray Imaging and Analysis, Department of Physics, University of Antwerp, 2000 Antwerp, Belgium; 3Lab for Equilibrium Investigations and Aerospace, Department of Physics, University of Antwerp, 2000 Antwerp, Belgium

**Keywords:** FRP, dark field, edge illumination, phase contrast imaging, constrained spherical deconvolution

## Abstract

The properties of fiber reinforced polymers are strongly related to the length and orientation of the fibers within the polymer matrix, the latter of which can be studied using X-ray computed tomography (XCT). Unfortunately, resolving individual fibers is challenging because they are small compared to the XCT voxel resolution and because of the low attenuation contrast between the fibers and the surrounding resin. To alleviate both problems, anisotropic dark field tomography via grating based interferometry (GBI) has been proposed. Here, the fiber orientations are extracted by applying a Funk-Radon transform (FRT) to the local scatter function. However, the FRT suffers from a low angular resolution, which complicates estimating fiber orientations for small fiber crossing angles. We propose constrained spherical deconvolution (CSD) as an alternative to the FRT to resolve fiber orientations. Instead of GBI, edge illumination phase contrast imaging is used because estimating fiber orientations with this technique has not yet been explored. Dark field images are generated by a Monte Carlo simulation framework. It is shown that the FRT cannot estimate the fiber orientation accurately for crossing angles smaller than 70${}^{\circ }$, while CSD performs well down to a crossing angle of 50${}^{\circ }$. In general, CSD outperforms the FRT in estimating fiber orientations.

## 1. Introduction

Fiber-reinforced polymers (FRPs) are a type of composite material, where fibers (carbon, glass, aramid, …) are embedded in a polymer matrix, resulting in a material that is strong and rigid, while simultaneously being lightweight [1,2]. Thanks to these properties, FRPs are desirable in industries such as aerospace [3,4], automotive [5,6], building and construction [7,8,9] and sports [3]. The unique characteristics of FRPs strongly depend on the orientation distribution of the fibers inside the polymer matrix and how the fibers are woven together [10,11,12,13]. Thus, it is vital to obtain information about the fiber orientations to assess the quality of a material component [14].

X-ray computed tomography (XCT) is a well-known non-destructive imaging technique to visualize the internal structure of composite materials. Unfortunately, XCT suffers from two challenging aspects concerning the visualization of fibers. Firstly, conventional XCT relies only on attenuation contrast to visualize structures within the scanned object. The fibers and the polymer matrix in which they are embedded have similar attenuating properties, causing the contrast between them to be low [15]. Secondly, the fibers must be resolved individually to extract the orientation, which can be challenging because fibers are typically only a few micrometers thick. Although high resolution micro-XCT has a sufficient resolution to distinguish individual fibers, it suffers from a limited field of view that spans only a few millimeters [15].

Both challenges can be overcome with X-ray phase contrast imaging (XPCI), which provides additional contrasts that are complementary to the attenuation contrast, namely phase contrast and dark field contrast, allowing to distinguish the fiber bundles from the polymer matrix [16]. Additionally, XPCI allows to determine the orientation of the fibers without the need to resolve them individually. Instead of acquiring high resolution images that suffer from a small field of view, in which each fiber is visualized individually, the fiber orientation can be determined from images with a lower resolution and consequently larger field of view. This is possible because the information captured by the dark field contrast is linked to ultra-small-angle X-ray scattering [17]. Indeed, when X-rays pass through a material with microstructures, the repeated refractions at the material interfaces cause the X-rays to scatter. As long as the microstructures do not have a specific directionality (e.g., grains or bubbles), the X-ray scattering is isotropic. However, microstructures such as fibers cause anisotropic X-ray scattering, allowing the extraction of the fiber orientations through dedicated reconstruction techniques.

Directional dark field imaging, is an early method that showed the variation of the dark field contrast in function of the rotation angle in 2D [18]. Later, Malecki et al. developed a method for grating based interferometry (GBI) to reconstruct the anisotropic dark field signal in 3D, called X-ray tensor tomography (XTT) [19]. In XTT, the scatter magnitude is reconstructed in a number of predefined directions distributed over the unit sphere. Then, for each voxel, an ellipsoid is fitted to the set of reconstructed scatter magnitudes, describing the scatter tensor in that voxel. Finally, the fiber orientation is given by the smallest principal axis of the ellipsoid, under the assumption that the scatter is minimal along the fiber orientation and maximal orthogonal to the fiber orientation. Following this research, Vogel et al. developed constrained X-ray tensor tomography (CXTT) by generalizing the reconstruction scheme to allow additional constraints and replacing the ellipsoid fitting by principal component analysis (PCA) to extract the direction of minimal scatter (and thus the fiber orientation) [20]. However, the downside of the PCA approach is that only one fiber direction can be extracted per voxel. In response to this, Wieczorek et al. developed anisotropic dark field tomography (ADFT), where spherical harmonics represent the local scatter function, allowing the detection of multiple crossing fiber orientations within a voxel [21]. This is done by transforming the local scatter function with a Funk-Radon transform (FRT) to obtain an orientation density function (ODF). Finally, the fiber orientations in each voxel are obtained by extracting the local maxima of the ODF [22]. However, the ODF resulting from the FRT suffers from low angular resolution and introduces a bias on the fiber orientations when the fiber crossing angle is smaller than $90^{\circ }$ [23].

While prior work focused on GBI, in this work, we explore the use of edge illumination XPCI (EI-XPCI) to acquire simulated dark field images and estimate fiber orientations. EI-XPCI allows for more relaxed requirements on the spatial coherence of the source compared to GBI, enabling the use of uncollimated focal spots [24]. The dark field images are simulated via a Monte Carlo simulation framework in which the EI-XPCI setup is modeled. Then, the scatter magnitude is reconstructed with CXTT in a number of directions for each voxel. However, instead of relying on the FRT to retrieve multiple fiber orientations per voxel, we extract the orientations through constrained spherical deconvolution (CSD) [25], building upon our preliminary work [26]. A comparison is made between the FRT and CSD, where these techniques are evaluated in their capability to retrieve the correct fiber orientations and estimate the fiber crossing angles. We show that the FRT allows to accurately estimate the fiber orientations down to a fiber crossing angle of 70${}^{\circ }$, while CSD remains accurate down to a crossing angle of 50${}^{\circ }$.

## 2. Materials and Methods

### 2.1. Edge Illumination X-ray Phase Contrast Imaging

Edge illumination XPCI (EI-XPCI) is a phase contrast imaging technique that employs two absorbing masks with parallel aperture slits to introduce sensitivity for three complementary contrasts: attenuation, phase and dark field contrast [24]. The sample mask is placed in front of the sample and splits the X-ray cone beam emitted by the source into multiple smaller beams, called beamlets. The beamlets traverse through the sample and towards the detector, each beamlet corresponding to one pixel. In front of the detector, a detector mask is placed such that the apertures coincide with the center of each pixel, while the pixel edges are covered. Then, to extract the complementary contrasts, the sample mask is shifted multiple times parallel to the detector plane, as shown in Figure 1a, and an image is acquired at each sample mask position. This so-called phase stepping of the sample mask causes the measured intensity to vary with the position of the beamlet in each pixel. The resulting relation between the measured intensity and the position of the sample mask is characterized by the illumination curve (IC), which is typically modeled as a Gaussian function. When a sample is introduced behind the sample mask, the beamlets will interact with the material, leading to attenuation, refraction and scatter. In this way, the surface area under the IC is reduced, the IC peak position shifts and the IC broadens, as illustrated in Figure 1b. Finally, the three contrasts can be extracted from a pixel-by-pixel based comparison between the IC with and the IC without sample [27].

### 2.2. Simulating the Dark Field

Dark field images are obtained with simulations in GATE [28,29], a Monte Carlo simulation framework for tomographic emission, based on Geant4 [30]. Additionally, GATE was augmented with functionalities that are vital for phase contrast imaging [31]. In GATE, each photon emitted by the source interacts with the object through different physical processes, before reaching the detector.

All computations in GATE are CPU-based, which makes it in general a slow process. Despite CPU-based parallelization, extensive simulations can take multiple hours to complete, which is problematic for EI-XPCI simulations because the simulation must be repeated multiple times to simulate the phase stepping of the sample mask, as discussed in Section 2.1. A possible way to overcome this issue, is by using a virtual gratings approach for the phase stepping [32]. However, handling the large data files produced by the virtual gratings ROOT output can be challenging for complex simulations. Alternatively, it is possible to perform the phase stepping by shifting the detector mask simultaneously with the detector, while keeping the sample mask stationary during the simulation. Although this does not suffice to alleviate the problem, our augmented GATE version provides a feature that allows subsampling of the detector plane, called a virtual detector grid. This feature was originally added to simulate wave front generation for GBI [31], but can also be used to apply the effect of the detector mask post Monte Carlo simulation, similar to the virtual gratings approach. As a consequence, the detector mask can be removed from the simulation and phase stepping is performed post-simulation through virtually shifting the detector mask and detector pixel locations. In practice, this is done by applying a binary mask on the high resolution grid images, representing the detector mask. This approach reduced the simulation time with a factor of 5.

### 2.3. The Acquisition Scheme

The dark field signal is linked to the X-ray scattering properties of the microstructures inside a voxel. The magnitude of this scattering is anisotropic because it depends on the direction of the X-rays relative to the fiber orientations. Additionally, the scatter magnitude varies depending on the sensitivity direction of the phase contrast imaging setup. With EI-XPCI, the sensitivity direction is orthogonal to the aperture slits and is thus determined by the orientation of the masks. That is, the dark field signal is a function of both the beam direction and the sensitivity direction of the EI-XPCI setup [33]. In our simulation study, we focussed on the simplest acquisition scheme discussed by Sharma et al. [34]. We generated a set of 50 sensitivity directions, distributed on the full unit sphere, based on a bipolar electrostatic repulsion model [35]. Then, each sensitivity direction was sampled with 100 beam directions, resulting in a total of 5000 projection angles. Due to the nature of the EI-XPCI setup, the beam direction is always orthogonal to the sensitivity direction. Hence, the beam directions were distributed equally spaced on a circle in the plane orthogonal to each sensitivity direction. In practice, this results in a sequence of standard circular CT-trajectories, one for each sensitivity direction, wherein the rotation axis of the sample is parallel to the sensitivity direction. The generated sensitivity directions and corresponding beam directions are shown in Figure 2. The symmetry of the dark field signal can be used to generate additional data, since the dark field signal corresponding to a certain sensitivity direction is equal to the dark field signal of the opposite sensitivity direction, resulting in a set of 100 sensitivity directions in total.

### 2.4. Reconstructing Anisotropic Dark Field Images

Dark field images acquired with EI-XPCI cannot be reconstructed in the same way as attenuation images, where a single scalar is sufficient to describe the X-ray attenuation in a voxel of the reconstructed volume. Rather than reconstructing a single dark field scalar, multiple scatter magnitudes are reconstructed per voxel with CXTT [20]. Here, the scatter magnitude is reconstructed for a set of scatter directions that are distributed over the unit sphere. The number and orientation of the scatter directions can be freely chosen, but should be well distributed. Therefore, to optimize the reconstruction, the directions are distributed using a bipolar electrostatic repulsion model, similarly to the acquisition scheme in Section 2.3. However, the set of directions for which the scatter magnitude is reconstructed does not need to be the same as the set of directions in which the dark field signal was sampled. Here, we chose to reconstruct the scatter magnitude for 70 scatter directions, distributed over the full unit sphere.

Alternatively, it is also possible to reconstruct each circular CT-trajectory separately to yield a scatter magnitude in each direction, since the sensitivity direction is parallel to the rotation thus constant during rotation, similar to [36].

### 2.5. Estimating the Fiber Orientations

In the work of Wieczorek et al. [21], multiple crossing fiber directions are extracted by first performing ADFT, which results in a set of spherical harmonic coefficients that describe the scatter function in each voxel. Then, a FRT is used to transform the local scatter function to an ODF that contains the orientation information of the fibers. Assuming that the scattering is largest orthogonal to the main fiber orientation, the fiber orientations can be extracted from the local maxima in the ODF. However, the FRT is merely a reshaping of the scatter function and does not decouple the orientation information from the scatter information.

In order to achieve this decoupling of information, we performed a constrained spherical deconvolution on the scatter magnitudes in each voxel, reconstructed by 100 CXTT iterations. Spherical deconvolution is a technique developed by Tournier et al. to estimate fiber orientations from diffusion-weighted MRI data [37]. In this technique, the measured scatter function S(θ,ϕ) is modeled as a spherical convolution of the fiber orientation density function (ODF) F(θ,ϕ) with a response function R(θ): (1)S(θ,ϕ)=F(θ,ϕ)⊗R(θ),
where θ and ϕ represent the spherical coordinates, ⊗ denotes the spherical convolution symbol and R(θ) is estimated from voxels with a parallel oriented fiber population. The ODF F(θ,ϕ) represents the fraction of fibers in the voxel along the direction (θ,ϕ) and is obtained by spherically deconvolving the measured scatter function S(θ,ϕ) with the response function R(θ) [37]. Since spherical deconvolution is an ill-conditioned problem, constraints are added to avoid negative amplitudes in the ODF, which is known as constrained spherical deconvolution (CSD). These constraints enable to estimate more parameters than the number of measured scatter directions, resulting in an ODF with a larger angular resolution that is represented with a higher maximal degree of spherical harmonics, also know as super-resolved CSD [25]. When extracting the fiber orientation through CSD, the assumption that the scatter is the largest orthogonal to the main orientation of the fiber is avoided. Indeed, the scatter behavior is encompassed in the response function extracted from a region with parallel fibers.

The (super-resolved) CSD is directly performed on the set of reconstructed scatter magnitudes using MRtrix3, an open source software package for image processing and visualization, mainly oriented to diffusion MRI [38]. First, the response function is estimated from the region of parallel fibers with the *dwi2response* command. Then, the data set of 70 scatter magnitudes per voxel is spherically deconvolved with the response function, using the *dwi2fod* command with the single-shell single-tissue algorithm [25]. This results in a set of 120 spherical harmonic coefficients per voxel, describing the ODF up to a spherical harmonic degree of 14.

The FRT is calculated from the same data set of scatter magnitudes by first converting the magnitudes to spherical harmonic coefficients with the *amp2sh* command. This results in a set of 66 coefficients per voxel, describing the scatter function up to a spherical harmonic degree of 10. Then the FRT is performed on the set of coefficients, as explained in [39], yielding a second set of 66 coefficients that describe the ODF in each voxel.

Once the ODFs are obtained, either by FRT or CSD, the fiber orientations are extracted by finding the local maxima of the ODF with a peak finding algorithm in MRtrix3 [40]. Using the *sh2peaks* command, two fiber orientations are extracted per voxel in the form of a pair of vectors. The length of these vectors scales with the amplitude of the corresponding ODF. A threshold of 10% of the amplitude of the largest peak is introduced to prevent the algorithm from extracting faulty fiber orientations corresponding to small peaks that may occur in the ODF.

### 2.6. Experiments

The EI-XPCI setup was modeled after the FleXCT, a flexible XCT-scanner available at imec-Vision Lab. Absorbing masks were specifically designed for this imaging setup [41]. Without loss of generalization, only a small part of the detector (30 × 30 pixels) was modeled to limit the computation time of the simulations. The pixels had a size of 150 μm×150 μm, resulting in a detector plane of 4.5 mm×4.5 mm, positioned at a distance of 1798.5 mm from the source. The sample mask had an aperture width of 20  μm and a period of 100 μm, a thickness of 225 μm and was positioned at a distance of 1199 mm from the source, resulting in a magnification factor of 1.5. The sample mask was made of an impermeable material. As discussed in Section 2.2, there was no detector mask defined in the simulation and a high-resolution grid of 450×450 grid pixels was used instead. In this way, each detector pixel is sub-sampled by 15×15 grid pixels with a size of 10 μm×10 μm. The aperture width of the detector mask, applied in post processing, was 30 μm. In total, 5 phase steps were performed where the mask was shifted to the positions [−20 μm,−10 μm,0 μm,10 μm,20 μm] with respect to its centered position. The cone beam source emitted monochromatic X-rays with an energy of 25 keV.

The sample consisted of an epoxy sphere (1.4 mm radius) with a density of 1 g ${\mathrm{cm}}^{-3}$. The epoxy sphere enclosed three cubic shaped regions (0.5 mm wide) that were filled with carbon fibers that had a diameter of 4  μm and a density of 2.1 g ${\mathrm{cm}}^{-3}$. Fiber positions were generated using Poisson disk sampling to ensure a tightly packed randomized distribution, while avoiding overlapping fibers [42]. The epoxy sphere and the three fiber regions are depicted in Figure 3, where the blue fiber region in the center of the sphere contained 2403 parallel fibers in the *z*-direction with a length of 500 μm, equal to the side of the cubic region. The red and green fiber regions contained crossing fibers. Half of the fibers pointed in the *z*-direction. The other half of the fibers were rotated around the *y*-direction to introduce a crossing angle. However, to ensure that the crossing fibers did not overlap, each voxel of the region was divided in two sub-voxel sectors of 100 μm×50 μm×100 μm, each containing only fibers of the same orientation. Therefore, the fibers were shorter, with a length of 90 μm. In total, there were approximately 13,000 fibers in each cubic region with crossing fibers.

### 2.7. Analysis

The FRT and CSD are compared with respect to their capability to estimate fiber orientations and the angle between crossing fiber populations. The workflow is depicted in Figure 4. First, all the imaging parameters were given to GATE in which the simulation was performed. As output, GATE yielded high-resolution images that were converted to phase-stepped images, as discussed in Section 2.2. Then, phase retrieval was performed on the images to retrieve the three complementary contrasts, resulting in 5000 dark field images of 30×30 pixels. Next, these dark field images were reconstructed with 100 CXTT iterations to obtain the scatter magnitude for a set of 70 scatter directions in a volume of 30×30×30 voxels (100 μm voxel size). Then, the ODF for each voxel is obtained by a FRT and CSD, where the maximal spherical harmonic degree of the ODF obtained by FRT and CSD was 10 and 14, respectively. The required response function was estimated by MRtrix3 from the region of parallel fibers in the center of the epoxy sphere. Finally, the local maxima in the ODF, which indicate the orientations of the fibers, are retrieved by a peak finding algorithm, as explained in Section 2.5.

To compare both methods quantitatively, the fiber orientations were retrieved for a region of interest of 3×3×3 voxels within each crossing fiber region in the sample (avoiding the edges). In total, three separate simulations were performed, where each simulation contained fibers with different crossing angles, yielding six fibrous regions with the following crossing angles to compare: $\left[90{}^{\circ },80{}^{\circ },70{}^{\circ },60{}^{\circ },50{}^{\circ },40{}^{\circ }\right]$.

## 3. Results

### 3.1. Comparison between the Attenuation and Dark Field Contrast

To illustrate the complementarity of attenuation and dark field contrast generated by the fibrous regions, the 5000 dark field images were reconstructed with the simultaneous iterative reconstruction technique (SIRT) [43], ignoring the directional dependency of the dark field. The dark field reconstruction is compared to the SIRT reconstruction of the attenuation images, shown in Figure 5. Here, the increased contrast between the fibrous regions and the epoxy sphere is clearly visible in the dark field reconstruction of the object, while in the attenuation reconstruction the fibrous regions are difficult to distinguish.

### 3.2. Visual Comparison between the ODFs Calculated by the FRT and CSD

The ODFs calculated by a FRT and CSD are shown in Figure 6, for a slice through the center of the crossing fiber regions of the sample volume. The fiber populations in the top-left region had a crossing angle of 90${}^{\circ }$, while the fibers in the bottom region had a crossing angle of 70${}^{\circ }$. The ODFs are colored by orientation, where red corresponds to the *x*-direction, green the *y*-direction and blue the *z*-direction.

Additionally, individual ODFs calculated by the FRT and CSD, are shown in more detail in Figure 7 for a fiber crossing angle of 90${}^{\circ }$, 70${}^{\circ }$ and 40${}^{\circ }$.

### 3.3. Comparison of the Estimated Fiber Orientations

The fiber orientations extracted by the peak finding algorithm from the ODFs calculated with the FRT and CSD, are shown in Figure 8 and Figure 9 for the different crossing angles. In each figure, the center slice through the region with crossing fibers is shown. The color of the vectors in each voxel corresponds to the orientation, where red stands for the *x*-direction, green the *y*-direction and blue the *z*-direction.

In Figure 10, the angular deviations from the true fiber orientations in the region of interest are shown for different fiber crossing angles. Each boxplot contains the angular deviations of both crossing fiber orientations of each voxel, where the outliers are indicated by a ‘+’.

### 3.4. Comparing the Estimated Fiber Crossing Angle to the True Fiber Crossing Angle

Additionally, the angle between both estimated orientations is calculated and plotted for different crossing angles in Figure 11. The corresponding median angles and median absolute deviations (MAD) are listed in Table 1, together with the percentage of voxels in which the peak finding algorithm could not detect two separate fiber orientations. This occurs when the amplitude of the peak is below the specified threshold, consequently mislabeling the voxels as containing only parallel fibers or no fibers at all.

## 4. Discussion

The regions of fibers, shown in Figure 5, are hard to distinguish in the attenuation reconstruction, because epoxy and carbon have a similar density. However, the fibrous regions are clearly visible in the dark field reconstruction. Only the regions with fibers and the edge of the epoxy sphere produce dark field contrast. Because the epoxy sphere is homogeneous, the beamlets traversing the sphere will not broaden through X-ray scattering and thus will not cause any dark field contrast. In the SIRT reconstruction, the directional dependency of the dark field signal was ignored. Thus, while resulting in a high contrast between the fibrous and non-fibrous regions, the directions of the fibers cannot be estimated from this reconstructed image.

In Figure 6, the ODFs retrieved through FRT and CSD are shown for a slice through the center of the fibrous regions in the epoxy sphere. Individual ODFs, calculated by FRT and CSD, are shown for different fiber crossing angles in Figure 7. Here, it can be clearly seen that the ODF retrieved by CSD has a much higher angular resolution compared to the ODF calculated with a FRT, which looks blurred. Consequently, when the crossing angle is smaller, it becomes more difficult for the peak finding algorithm to find the correct orientations of the fibers because the peaks start overlapping. At a crossing angle of 40${}^{\circ }$, the peaks of the ODFs calculated with the FRT completely overlap, making it impossible to resolve the second peak.

The estimated fiber orientations are shown in Figure 8 and Figure 9 for the center slice through the fiber regions, where each voxel contained fibers in the *z*-direction and fibers rotated around the *y*-direction to introduce a crossing angle. For all these fibrous regions, a region of interest of 3×3×3 voxels was extracted for which the angular deviations from the true fiber orientations were calculated, as shown in Figure 10. Here, the capability of the FRT and CSD to estimate the correct fiber orientation are compared for different crossing angles. For a crossing angle of 90${}^{\circ }$ and 80${}^{\circ }$, both techniques have a similar performance. However, at a crossing angle of 70${}^{\circ }$, the fiber orientations estimated via the FRT become noticeably less accurate than the ones estimated trough CSD. This trend continues as the crossing angle gets smaller. At 50${}^{\circ }$ the FRT is incapable to estimate the correct fiber orientations, while the CSD still performs relatively well. As the fiber crossing angle decreases, the angular deviation of the fiber orientations estimated by CSD also increases, but remains low compared to the results obtained by the FRT.

These results are also reflected in the estimated crossing angles, shown in Figure 11. Here, the CSD outperforms the FRT in extracting the correct fiber crossing angle. The FRT performs well down to a crossing angle of 70${}^{\circ }$, below which the MAD increases significantly to 19.1${}^{\circ }$. In contrast, the crossing angles estimated through CSD are overall more accurate. However, for a crossing angle of 40${}^{\circ }$, even CSD does not estimate the angle accurately. Additionally, the peak finding algorithm does not always succeed in extracting two fiber orientations from the ODF calculated by the FRT, with up to 33% of the voxels being incorrectly assigned a single fiber orientation at a crossing angle of 60${}^{\circ }$. This also occurs for the ODFs calculated by CSD, although in lesser degree. This issue is directly linked to the value of the threshold in the peak finding algorithm, which has been empirically set to 10% of the amplitude of the largest peak. If the threshold is set to 0, the peak finding algorithm always finds two peaks in the ODF calculated by CSD, however these can be faulty orientations corresponding to small peaks around the origin of the ODF, that do not correspond to a physical fiber direction (visible on the bottom right in Figure 7). The threshold of 10% was chosen empirically, to remove the incorrect orientations, while retaining most of the true orientations.

In future work, the potential for machine learning methods could be investigated to improve the current method in the extraction of fiber orientations from dark field images. Although this simulation study was extensive, the value of physical experiments should not be disregarded. Therefore, future work should also cover real EI-XPCI scans of a larger object. In this study, the size of the object was limited to a few millimeters to decrease the simulation time.

## 5. Conclusions

In this work, we presented CSD as an alternative technique to the current FRT to estimate fiber orientations in composite materials using edge illumination based X-ray dark field tomography. The results show that CSD outperforms the FRT in estimating the correct fiber orientations as well as the inter-fiber crossing angles. The FRT is capable of accurately estimating the fiber orientation down to a fiber crossing angle of 70${}^{\circ }$, while the orientation estimation via CSD is accurate down to a crossing angle of 50${}^{\circ }$.

## Figures and Tables

**Figure 1 polymers-15-02887-f001:**
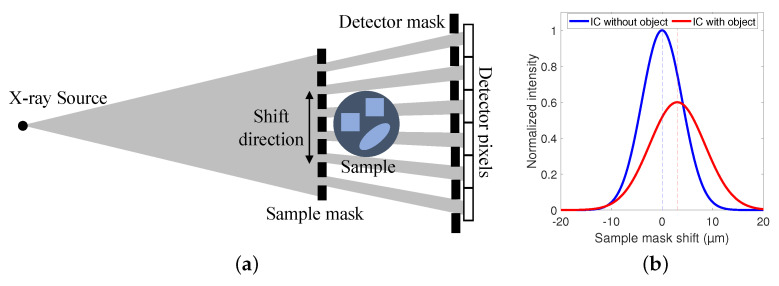
(**a**) A schematic representation of an edge illumination X-ray phase contrast imaging (EI-XPCI) setup. (**b**) An example illustrating the change between the illumination curve (IC) with and without object.

**Figure 2 polymers-15-02887-f002:**
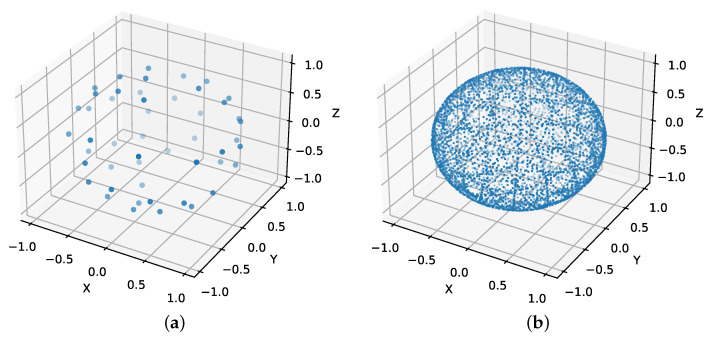
(**a**) The set of 50 sensitivity directions generated on the unit sphere by a bipolar electrostatic repulsion model. (**b**) The beam directions corresponding to the set of sensitivity directions.

**Figure 3 polymers-15-02887-f003:**
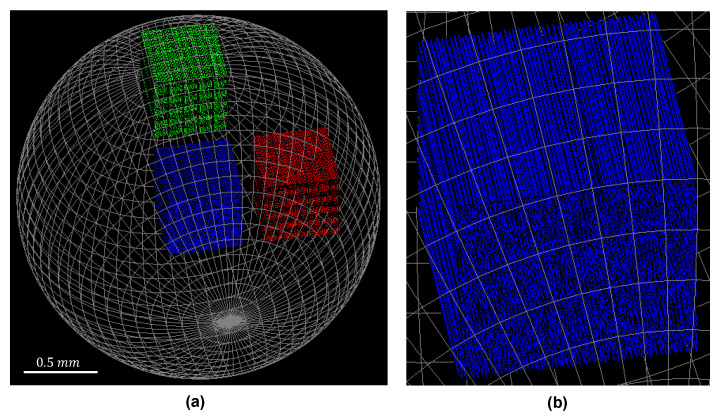
A visualization of the sample in GATE: (**a**) The epoxy sphere is shown as a gray wire frame, containing the three cubic fiber regions. (**b**) The blue fiber region in the center of the epoxy sphere, containing parallel carbon fibers.

**Figure 4 polymers-15-02887-f004:**
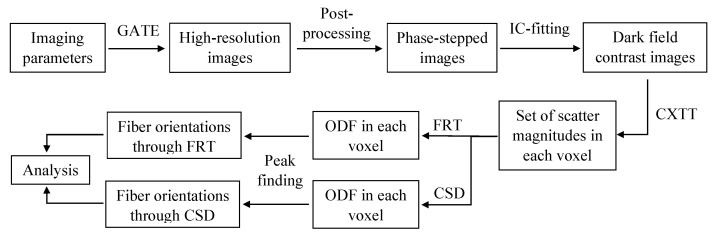
Workflow used to compare the fiber orientations retrieved with the Funk-Radon transform (FRT) and constrained spherical deconvolution (CSD).

**Figure 5 polymers-15-02887-f005:**
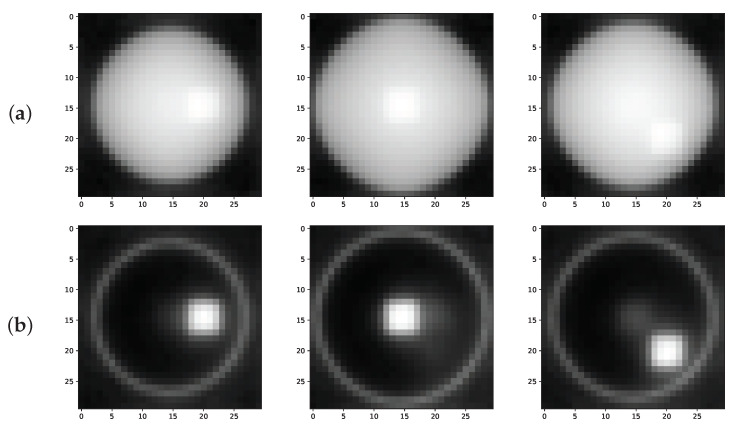
Slices at a different height through a SIRT-reconstruction of the epoxy sphere, showing the fibrous regions with: (**a**) attenuation contrast (**b**) dark field contrast.

**Figure 6 polymers-15-02887-f006:**
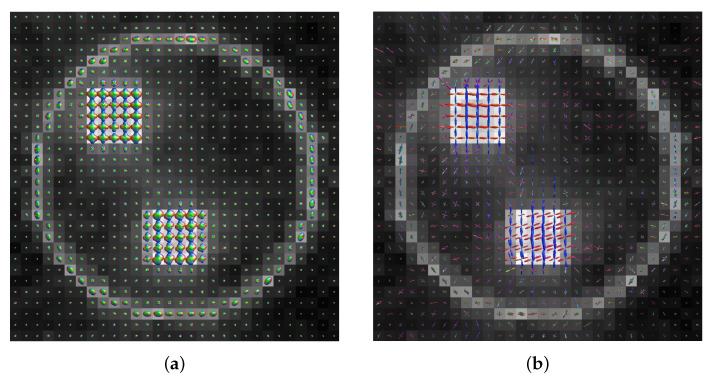
A slice through center of the crossing fiber regions in the sample. The top-left fibrous region contained fibers crossing at 90${}^{\circ }$, while the bottom region contained fibers crossing at 70${}^{\circ }$. In each voxel the orientation density function (ODF) is shown, retrieved by the (**a**) FRT or (**b**) CSD.

**Figure 7 polymers-15-02887-f007:**
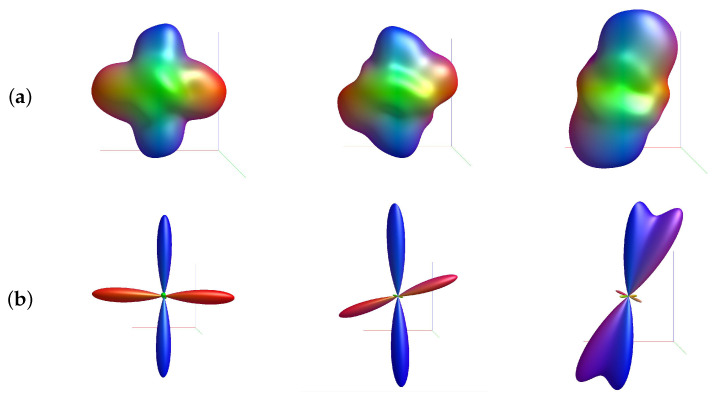
Individual ODFs retrieved by (**a**) FRT and (**b**) CSD for a crossing angle of (left-to-right) 90${}^{\circ }$, 70${}^{\circ }$ and 40${}^{\circ }$.

**Figure 8 polymers-15-02887-f008:**
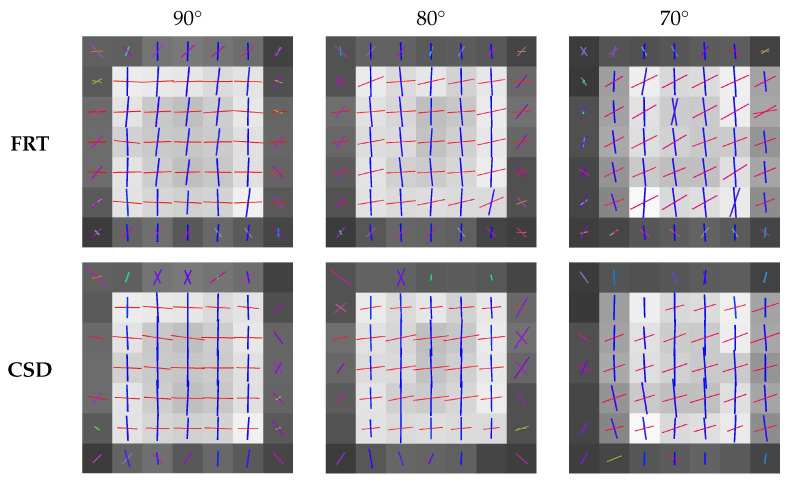
Slices through the center of the fibrous regions with estimated fiber orientations crossing at 90${}^{\circ }$, 80${}^{\circ }$ and 70${}^{\circ }$, extracted through the FRT and CSD.

**Figure 9 polymers-15-02887-f009:**
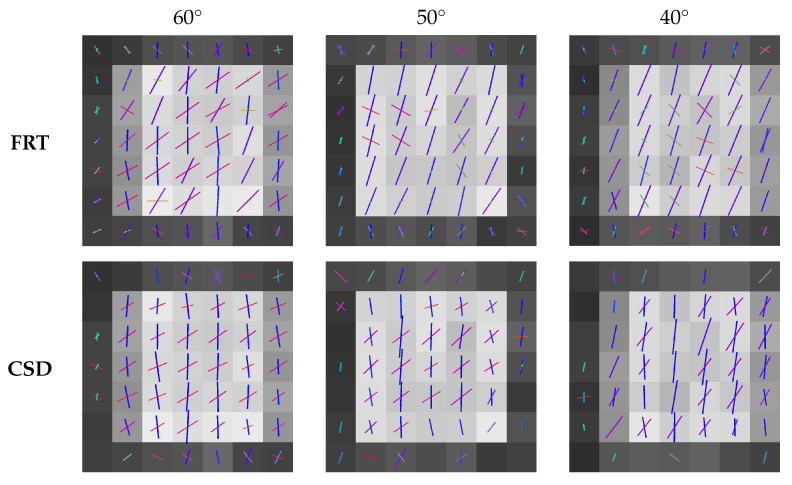
Slices through the center of the fibrous regions with estimated fiber orientations crossing at 60${}^{\circ }$, 50${}^{\circ }$ and 40${}^{\circ }$, extracted through the FRT and CSD.

**Figure 10 polymers-15-02887-f010:**
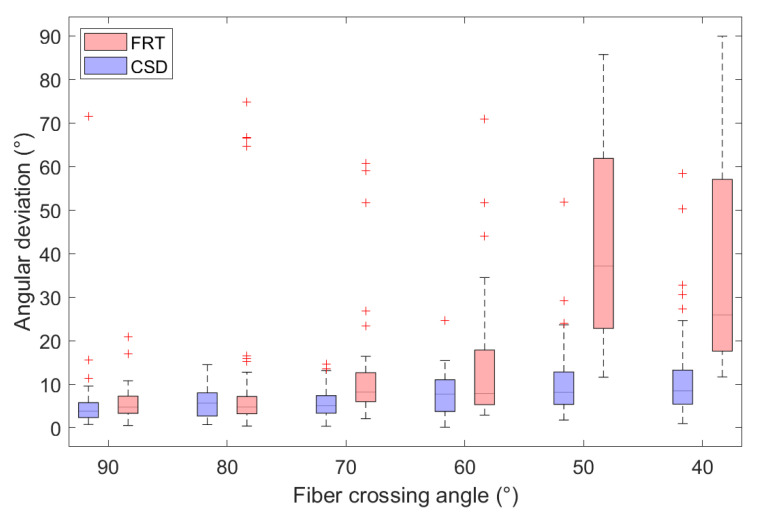
Comparison of the angular deviation between the estimated and true fiber orientations, retrieved through a FRT and CSD for various crossing angles. Outliers in the boxplots are indicated by ‘+’.

**Figure 11 polymers-15-02887-f011:**
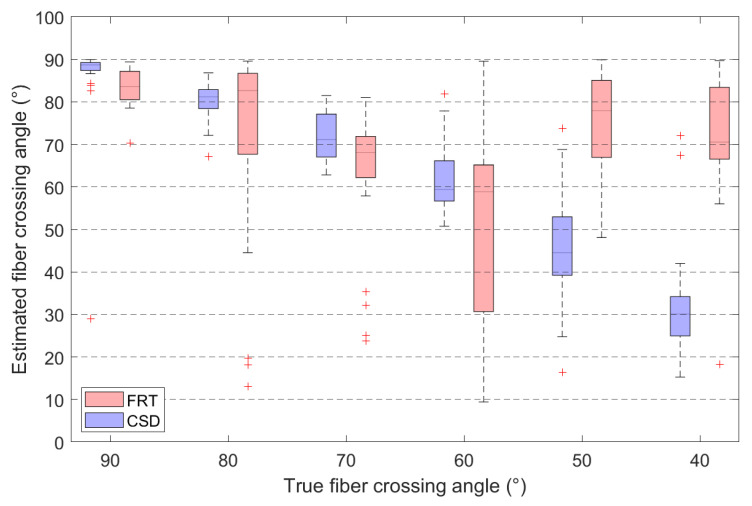
Comparison between the estimated fiber crossing angles retrieved through the FRT and CSD, for various crossing angles. Outliers in the boxplots are indicated by ‘+’.

**Table 1 polymers-15-02887-t001:** The median of the estimated fiber crossing angles, the median absolute deviation (MAD) and the percentage of voxels in which no crossing fibers were detected (mislabeled), corresponding to the data visually represented in Figure 11.

	FRT			CSD		
Crossing Angle	Median	MAD	Mislabeled	Median	MAD	Mislabeled
90${}^{\circ }$	83.6 ${}^{\circ }$	3.3 ${}^{\circ }$	0%	88.7 ${}^{\circ }$	0.9 ${}^{\circ }$	0%
80${}^{\circ }$	82.7 ${}^{\circ }$	6.0 ${}^{\circ }$	0%	81.2 ${}^{\circ }$	2.2 ${}^{\circ }$	0%
70${}^{\circ }$	68.1 ${}^{\circ }$	4.6 ${}^{\circ }$	7%	71.1 ${}^{\circ }$	5.1 ${}^{\circ }$	0%
60${}^{\circ }$	58.9 ${}^{\circ }$	19.1 ${}^{\circ }$	33%	59.4 ${}^{\circ }$	3.4 ${}^{\circ }$	0%
50${}^{\circ }$	77.9 ${}^{\circ }$	7.9 ${}^{\circ }$	19%	44.5 ${}^{\circ }$	6.8 ${}^{\circ }$	0%
40${}^{\circ }$	70.5 ${}^{\circ }$	7.7 ${}^{\circ }$	22%	30.1 ${}^{\circ }$	4.5 ${}^{\circ }$	15%

## Data Availability

The data presented in this study are openly available in Zenodo at https://doi.org/10.5281/zenodo.8089557 (accessed on 23 June 2023)

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
