# Peer review of "Fiber Orientation Estimation from X-ray Dark Field Images of Fiber Reinforced Polymers Using Constrained Spherical Deconvolution"

_polymers, 2023, doi:10.3390/polym15132887_

Round 1
Reviewer 1 Report
The paper proposes constrained spherical deconvolution (CSD) as an alternative to the FRT to resolve fiber orientations. Monte Carlo simulation framework was used to generate the Dark field images. This paper is well originated with nice Figures and the research content is innovative. However, the authors are encouraged to consider the following comments for necessary improvement.
1. Abstract:
(1) A brief research background of orientation of fibers should be added in the first sentence of abstract.
(2) Rephrase the sentence “we employ……”, “We show that……” with “this paper employ……”, “The results show that……”.
(3) Rephrase the key words with “FRP, fiber orientation, dark field, edge illumination, phase contrast imaging, constrained spherical deconvolution”.
2. Introduction:
(1) As authors stated: FRPs are desirable in industries such as aerospace, automotive, building and construction and sports, this was because FRPs is lightweight, higher mechanical and superior aging and fatigue resistance. Please provide some latest research findings related to these discussions. The following relevant studies can be reviewed to make necessary supplements in the research background, such as Engineering Structures, 2023, 274: 115176. Composite Structures, 2019, 229: 111427. Polymers, 2023, 15:2483.
(2) The introduction should introduce more predecessors' research work on fiber orientation.
3. Materials and methods
(1) Whether the method in this paper has a requirement for the resolution of the picture?
(2) Whether the orientation of fibers can be distinguished through the post-processing of XCT scan images by machine learning method?
4. Results and Discussion
(1) What is the reason of the unrecognizable of fiber orientation under small angle?
(2) Further improvement of the current model should be given.
5. Conclusions
The conclusions should be condensed, key conclusions and findings should be given to highlight the innovation of the paper.
It needs the minor check and revision.
Author Response
Please see the attachement.

Reviewer 2 Report
The manuscript, “Fiber Orientation Estimation from X-ray Dark Field Images of Fiber Reinforced Polymers using Constrained Spherical Deconvolution: A Simulation Study” by, B. Huyge et al. demonstrates constrained spherical deconvolution (CSD) as an alternative technique to the current Funk-Radon transform (FRT) to estimate fiber orientations in composite materials using X-ray darkfield imaging. The proposed CSD method enables one to determine a crossing angle between fibers more accurately than the FRT method. This manuscript is suitable for publication in Polymers. Comments and questions to authors are in the followings;
- Authors described that all computations in GATE are CPU-based so that this makes the process slow and employed new ideas to overcome this issue. How much did authors improve simulations? It should be better to mention in the text.
- It seems that the amplitude threshold is a very important parameter to determine a cross angle between fibers and authors chose 10% for this research. Is there any special reason for selecting 10%? I am wondering if authors tried to calculate the angle with different threshold values.
- There are red crosses in Fig. 10 and 11. Authors need to explain what they are in the text.
Round 2
Reviewer 1 Report
The authors has provided a good response to the reviewer's comments, but no relevant modifications have been found in the revised manuscript, especially for the discussion and analysis. Can the authors provide the revised manuscript with a marked version?
Author Response
Dear reviewer,
We apologize for not including the marked version in the previous submission.
We now provide a marked version that shows the modifications made to the original manuscript.
Round 3
Reviewer 1 Report
Accepted.